# Life Conditions as Mediators of Welfare State Effect on Mental Wellbeing among Oldest Old in Europe

**DOI:** 10.3390/ijerph19074363

**Published:** 2022-04-05

**Authors:** Johanna Cresswell-Smith, Kristian Wahlbeck, Jorid Kalseth

**Affiliations:** 1Faculty of Medicine, University of Helsinki, P.O. Box 63, 00014 Helsinki, Finland; 2Mental Health Team, Finnish Institute for Health and Welfare, P.O. Box 30, 00271 Helsinki, Finland; kristian.wahlbeck@thl.fi; 3SINTEF Digital, P.O. Box 4760, 7465 Trondheim, Norway; jorid.kalseth@sintef.no

**Keywords:** mental health, wellbeing, public policy, social determinants, European Social Survey

## Abstract

Background: Mental wellbeing is formed by our daily environments, which are, in turn, influenced by public policies, such as the welfare state. This paper looks at how different aspects of life conditions may mediate the welfare state effect on mental wellbeing in oldest old age. Methods: Data were extracted from Round 6 of the European Social Survey (2012). The dataset comprised of 2058 people aged 80 years and older from 24 countries. Mediation analyses determined possible links between the welfare state, including eleven intervening variables representing life conditions and five mental wellbeing dimensions. Results: Our study confirms that the higher the level of welfare state, the better mental wellbeing, irrespective of dimension. Although several life conditions were found to mediate the welfare state effect on mental wellbeing, subjective general health, coping with income and place in society were the most important intervening variables. Conclusions: All three variables centre around supporting autonomy in the oldest old age. By teasing out how the welfare state influences mental wellbeing in the oldest old, we can better understand the many drivers of wellbeing and enable evidence informed age-friendly policy making.

## 1. Introduction

The oldest old population, defined here as adults aged 80 years and over [1] is projected to rise globally from 143 million in 2019 to 426 million by 2050 [2]. This demographic transition has stimulated broad discussions regarding the need for age-friendly environments [3]. Mental wellbeing is an important area of focus in this regard, with links to positive health outcomes [4], increased life expectancy [5] and economic benefits [6,7,8]. Our daily environments are shaped by the surrounding welfare state policies, which in turn, impact mental wellbeing also in oldest old age [9].

Definitions of mental wellbeing differ according to which distinct features and dimensions are emphasised [10]. Subjective wellbeing [11] approaches tend to emphasise the evaluative dimension of wellbeing with a focus on cognitive appraisals of life satisfaction, together with the hedonic or emotional dimension, which relates to a sense of pleasure and happiness [12,13,14]. Eudaimonic approaches, on the other hand, emphasise the importance of meaning, for example, via personal development and self-realisation. These approaches include Ryff’s definition of psychological wellbeing made up of life purpose, mastery and positive relatedness, flow, autonomy, personal growth and self-acceptance [15]. 

Keyes’ model [16,17] is built in relation to factors that influence flourishing and languishing, while Seligman’s PERMA model builds on five components, including positive emotions, engagement, relationships, meaning and accomplishment [18]. Approaching mental wellbeing as a multidimensional construct allows for a comprehensive understanding [19,20]. 

Exploring different dimensions of mental wellbeing sheds light on phenomenon, such as the ‘wellbeing paradox’, denoting a tendency for the evaluative dimension to increase in older age, particularly in the context of wealthier, English speaking countries [21]. Socio-economic or socio-political context appears to lie behind this paradox, as the effect has not been replicated in lower income countries or in post-communist countries [22]. Different aspects of wellbeing tend to vary according to age group and what dimensions of wellbeing are under consideration as well as on the individual characteristics and life circumstances of the study participants [23]. Furthermore, specific aspects of a country’s social policy have been found to influence wellbeing [23]. 

For example, high levels of emotional wellbeing (happiness) have been attributed to the influence of welfare state in Nordic countries [24] as have high levels of flourishing [25]. Wellbeing may be influenced by characteristics, such as low income inequality, high social trust and well-developed social welfare and health care systems [26]. These approaches parallel research on the importance of social determinants of mental health [27]. In line with this, the Mental Wellbeing Impact Assessment (MWIA) framework proposes four protective pathways for mental wellbeing—namely enhancing control, increasing resilience and community assets, facilitating participation and promoting inclusion [28].

Despite constituting a growing segment of the population, the oldest old age group tends to be overlooked in research [29,30]. In a step towards ameliorating this research gap, the European Welfare Models and Mental Wellbeing in Final Years of Life (EMMY) project (www.emmyproject.eu) used a mixed-methods approach to explore mental wellbeing in the oldest old age group. The EMMY project (2017–2019) made use of data from Round 6 of the European Social Survey (ESS). These included data from rotating modules on personal and social wellbeing and a broad range of items related to MWB [31]. 

Exploratory factor analysis [32] highlighted five important dimensions of mental wellbeing in oldest old age (1) the evaluative dimension capturing overall happiness and life-satisfaction; (2) the emotional dimension including enjoyment, calmness and happiness; (3) positive psychological functioning, built on autonomy, competence, self-esteem and optimism; (4) meaning and flow, describing states of presence and engagement; and (5) positive and supportive relationships that is, giving and receiving support and feeling appreciated [33]. A further paper originating from the EMMY project found that a welfare state mattered more for mental wellbeing in older age groups compared to younger age groups [9].

The current paper explores how the conditions of daily life influence the different dimensions of mental wellbeing in the oldest old age. More specifically we look at the mediating effect of these variables on the welfare state effect in terms of five dimensions of mental wellbeing. Teasing out the influence of these variables gives us a better understanding of the drivers of mental wellbeing in oldest old age, facilitating evidence to inform policy making for this growing population group.

## 2. Materials and Methods

### 2.1. The Source of Data and Variables

Data was extracted from Round 6 of the ESS (2012), which is collected on a broad array of topics via face-to-face interviews [31]. Data were restricted to individuals aged 80 years and older from 24 countries belonging to either the European Union or the European Free Trade Association. Both core and rotating ESS modules were included in this study—namely: Subjective Wellbeing, Social Exclusion, Religion, National and Ethnic Identity, Personal and Social Wellbeing, Media and Social Trust, Politics, as well as Socio demographics [31]. These modules measure different aspects of mental wellbeing, some of which are not available from later surveys waves. Specific items from these modules were compiled to form the mental wellbeing variables, as well as being used as intervening variables.

Mental wellbeing variables included the evaluative, emotional, psychological functioning, flow and relational dimensions [33] (see Table A1). The evaluative dimension of wellbeing refers to a sense of being satisfied with life and the evaluation of a positive affect, such as enjoyment, calmness and happiness. The emotional dimension also includes a feeling of vitality (i.e., being ‘full of energy’). The positive psychological functioning dimension refers to autonomy, competence, self-esteem and optimism. The flow dimension includes the perception of meaning, presence and engagement. The relational dimension consists of positive and supportive relationships, including receiving support and appreciation by others and helping and providing support to others [20]. The values for these dimensions were defined by calculating the average of the individual ESS items that make up a particular dimension. Individual items were first transformed (normalised) to a common scale using a min–max algorithm transforming each item into values between 1 and 5: New score = ((5–1)/(max-min)) * (score-max) + 5.

Intervening variables from the ESS were selected on the basis of their fit with the four MWIA protective factors, i.e., enhancing control, increasing resilience and community assets, facilitating participation and promoting inclusion (see Table A2).

### 2.2. Welfare State Index (WSI)

Esping-Andersen’s classification of welfare states by social rights and social spending, income redistribution and employment policy is the starting point for many studies investigating welfare state impacts [34,35,36]. This typology has been revised and expanded to include several countries and welfare state-typology variants [37]. However, these regime typologies do not allow for the study of possible variation within these regime types. In line with this, a more flexible composite WSI approach has been used [9] representing different aspects of welfare culture, welfare institutions, policy instruments and outcomes of welfare states [38,39]. 

The WSI includes the per capita Gross Domestic Product (GDP), health and social expenditures as a percentage of GDP, a gender empowerment measure, life expectancy at 65, work life duration and the Gini coefficient—a measure of individualism and a measure of social trust (see Table A3). The WSI was constructed using factor analysis restricted to one dimension, allowing for different weights for the eight variables. The predicted value of this factor was then used to represent the WSI in the analyses. The WSI correlates highly with the welfare state regime typology as illustrated in our previous study of wellbeing [9].

### 2.3. Statistical Analyses

Mediation analyses were applied in order to determine possible links between WSI, intervening (mediator) variables and mental wellbeing dimensions, as depicted in Figure 1.

Multilevel analysis were used to analyse the WSI impact on mental wellbeing, taking into account potential unobservable effects applying to individuals from the same country, as well as consequences of the welfare state variable being measured at the country level. Multilevel Structural Equation Modelling (MSEM) was employed to explore potential mediators [40].

#### The following Steps Were Performed

(1)An initial analysis of the total effect (path c in Figure 1) of WSI on mental wellbeing entailed five separate regression analyses, one for each mental wellbeing dimension, controlling for age and gender.(2)Mediating effects. An intervening variable is considered to be a significant mediator of welfare state effects (i.e., an indirect effect) if it is significantly associated both with WSI (path a in Figure 1) and with mental wellbeing (path b in Figure 1). If the indirect effect (a×b in Figure 1) is equal to zero, this indicates that the direct effect (c’ in Figure 1) equals the total effect (c). The magnitude of the mediation effect can be assessed by comparing coefficients of path c and c’ in Table 3.

MSEM is made up of simultaneous regression of (1) WSI on the intervening variable (path a in Figure 1) and (2) WSI (path c’ in Figure 1) and the mediator variable (path b in Figure 1) on mental wellbeing, taking into account the multilevel structure of the data, i.e., individuals being nested within a country. The product of the coefficients of paths a × b (in Figure 1) was used to test for indirect effects. The effects of all eleven intervening variables were analysed separately meaning a total of 11 * 5 MSEM regression analyses were performed, one for each intervening variable and each mental wellbeing dimension (including control for age and gender).

An intervening variable is deemed a mediator if a significant indirect effect is found, i.e., if the *p*-value is less than 0.05. A full mediation effect is present when the direct WSI effect (c’ in Figure 1) drops to zero. A partial mediation effect is found if the indirect effect (a × b in Figure 1) is statistically significant, and the direct WSI effect remains significantly different from zero.

All analyses were performed using Stata/SE 16.1. The mixed command was used to analyse the total effect of WSI on mental wellbeing, while the gsem and the postestimation nlcom commands were used in the MSEM regression and to test for indirect effect.

## 3. Results

### 3.1. Descriptions

#### 3.1.1. Participants

The dataset comprised 2058 participants, 62% of which were female. The average age was 84.1 years (SD 3.67), and the oldest participant was 103 years of age.

#### 3.1.2. Welfare State Variable

WSI measured at the country level produced values ranging from −1.65 for Bulgaria to 1.69 for Norway. By mapping the WSI scores (see Figure 2) a clear pattern was found with Nordic countries showing the highest WSI scores, followed by the Bismarckian countries, Anglo-Saxon countries, Southern countries and finally the Eastern European countries.

### 3.2. Intervening Variables

A total of 11 intervening variables (Table 1) were measured using different scales at the individual level; thus, the direct comparison of average scores across items is not possible.

### 3.3. Mental Wellbeing Variables

The normalised scores for individual mental wellbeing dimensions are shown in Table 2.

#### 3.3.1. Step 1

The first step in the analyses confirmed the total effect of WSI on mental wellbeing (path c in Figure 1) by way of separate regression analyses for each of the five dimensions controlling for age and gender. Figure 3 (left side panel) shows that the WSI variable is positively and significantly associated with all five dimensions of mental wellbeing. This implies that the higher level of WSI, the higher the level of mental wellbeing for each dimension (see right side panel of Figure 3). Higher levels of WSI correspond to high income countries with more well-developed welfare state mechanisms, in this case the Nordic countries and other north-western European countries.

Furthermore, the association of WSI was found to be highest for the evaluative dimension, followed by the emotional dimension, flow, psychological functioning and finally relational dimensions of mental wellbeing. Country differences show that scores for the evaluation dimension were considerably lower in countries with low WSI values compared to the relational dimension, while scores for the evaluative and relational dimensions were similar in countries with high WSI values.

#### 3.3.2. Step 2

The second step investigated whether the intervening variables mediated the association between WSI and individual dimensions of mental wellbeing. The coefficient for WSI without control for intervening variables (total effect c) is shown in the first row of Table 3. Coefficients with control for each intervening variable are listed underneath (direct effect c’). Further mediation diagrams can be found in Appendix B.

The indirect effect was analysed for all five mental wellbeing dimensions with WSI as the independent variable and the dimensions of mental wellbeing as dependent variables. A series of ESS items representing life conditions were used as intervening variables. All intervening variables, except for chance to show capabilities, were found to mediate a significant indirect effect on all mental wellbeing dimensions. The lack of significant indirect effect for the intervening variable chance to show capabilities was due to a lack of associations with WSI (see Table 1), as the association with mental wellbeing was strong (data not presented). Indirect effects by all other intervening variables were significant at *p* < 0.01.

Subjective general health, coping with income and place in society produced the largest difference between the c and c’ paths in the mediation model and will therefore be presented in more detail. These intervening variables ranked in the top three coefficients of the WSI (i.e., c–c’) for all mental wellbeing dimensions, with the exception of coping with income ranking fifth for the flow dimension. These three variables also had the highest association with the WSI (see Table 1).

The subjective general health item was obtained by asking participants to rate their own general health on a five-point Likert scale (European Social Survey, 2012) [31]. There was a significant positive relationship between WSI and subjective general health, the second highest association with WSI among all intervening variables (WSI effect = 0.411, *p* < 0.001, Table 1). The indirect effect for the subjective health variable was significant at *p* < 0.001 for all dimensions of mental wellbeing (see Table 3) meaning that controlling for subjective general health reduced the welfare state effect on all mental wellbeing dimensions considerably. 

Full mediation was found for psychological functioning and relational dimensions, although a strong partial meditation was also found for the other dimensions, for example the welfare state effect on the emotional dimension was reduced from 0.252 (c) to 0.093 (c’). Compared to the other intervening variables, the subjective general health item ranked as the strongest mediator for emotional, psychological functioning and flow dimensions as well as the second strongest mediator for the evaluative dimension and third for the relational dimension.

The coping with income variable was generated by asking participants about their current feelings about their household’s income ranging from “living comfortably” to “very difficult”. A significant positive relationship was found between WSI and coping with income, the highest association with WSI among the intervening variables (WSI effect = 0.574, *p* < 0.001, see Table 1), and the indirect effect for the coping with income variable was significant at *p* < 0.001 for all dimensions of mental wellbeing (see Table 3). Full mediation was found for the relational dimension, along with a strong partial meditation for other dimensions. The welfare state effect was for example reduced from 0.402 (c) to 0.232 (c’) on the evaluative dimension. Compared to the other intervening variables, the coping with income variable was the strongest mediator of welfare state effect in terms of the evaluative dimension, second strongest for the emotional and relational dimensions and third strongest for the psychological functioning dimension.

The place in society variable asked participants to categorise how they placed themselves in society on a scale from 0 to 10 with zero representing the bottom of society and ten representing the top of society [32]. There was a significant positive relationship between WSI and place in society, the third highest association with WSI among the intervening variables (WSI effect = 0.364, *p* < 0.001, Table 1), and the indirect effect for the place in society variable was significant at *p* < 0.001 for all dimensions of mental wellbeing. 

Full mediation was found for the relational dimension, and a strong partial meditation was found for the other dimensions. For example, the welfare state effect on the psychological functioning dimension was reduced from 0.147 (c) to 0.063 (c’). Compared to the other intervening variables, the place in society intervening variable had the strongest mediating effect in terms of the relational dimension; it was the second strongest mediator of the welfare state effect for the psychological functioning and flow dimensions and third for the evaluative and emotional dimensions.

## 4. Discussion

The results identify important mediators of the welfare state effect on five different dimensions of mental wellbeing in oldest old age. Our study confirms that higher levels of the WSI are associated with higher the levels of different dimensions of mental wellbeing. Furthermore, by exploring how the conditions of daily life mediate the welfare state effect in relation to these dimensions, three areas were found to be particularly influential: subjective general health, coping with income and place in society each impacted specific dimensions of mental wellbeing differently.

### 4.1. Subjective General Health and Mental Wellbeing

Health is especially important in oldest old age due to the higher likelihood of illness and increased support needs [41]. Older adults are often reliant on support, such as care and pension provisions for their wellbeing, which simultaneously serve to ameliorate health inequalities [42]. The current study found higher levels of subjective general health in stronger welfare states compared with less developed welfare states. The welfare state effect can therefore be considered to be significantly influenced by levels of subjective general health. 

Previous studies highlight the need for promoting health in order to improve life satisfaction, especially in economically disadvantaged population groups [43]. There is a positive correlation between subjective health and the emotional dimension of wellbeing. Declining health status in old age does not always produce a similar decline in the emotional dimension of mental wellbeing [21]. Despite the majority of people reaching oldest old age with multiple health difficulties, they still tend to report high levels of self-reported health, which may potentially be a reflection of psychological adaptation [44]. 

It remains unclear whether the association between this positive subjective assessment and the strength of the welfare state is due to stronger psychological adaptation, fewer health-related concerns due to better access to public services or some other contextual factor linked to the welfare state. Although the exact mechanisms are objects for further studies, the end result remains the same, which is higher levels of mental wellbeing among the oldest old in stronger welfare states. 

Similarly, the considerable influence of subjective health in relation to the psychological functioning dimension of mental wellbeing was demonstrated by the welfare state effect diminishing significantly (achieving full mediation) when subjective health was added to the mediation model. Positive subjective health appears to therefore support the psychological functioning dimension of mental wellbeing in oldest old age. Items making up the psychological functioning dimension have links to aspects of autonomy. Previous research also emphasised the importance of autonomy in older age, although highlighting that support may be necessary in accordance with age-related needs [45]. 

Therefore, welfare states that support health in oldest old age may simultaneously support a sense of autonomy. The flow dimension of mental wellbeing was also mediated by the subjective health variable and was higher in countries with stronger welfare state support. Supporting subjective health appears to therefore strengthen a feeling of meaning and engagement in oldest old age. Importantly, previous research underlined the ability derive enjoyment from daily activities despite health difficulties [46].

Subjective general health almost fully mediated the welfare state effect on the relational dimension. A good level of health was also found to be a protective factor against loneliness for people aged 60 years and over in a previous study [47]. Relationships are particularly important in oldest old age considering the higher risk of isolation and loneliness, for example, via the age-related losses of partners, siblings or friends [48].

### 4.2. Coping with Income and Mental Wellbeing

The basic purpose of the welfare state is to provide economic security and social equality, for example via social transfers, such as pensions. The current study found the welfare state effect on all dimensions of mental wellbeing to be mediated by the coping with income variable. Interestingly, the coping with income variable increased most with each incremental increase in WSI. Although research suggests that older adults report comparatively less financial difficulty than younger age groups, welfare state support appears to be especially important for individuals with higher support needs [49]. Income redistribution is one important aspect of welfare states; however, an equally important aspect of welfare states is access to non-income benefits and services.

The coping with income variable had the strongest impact on the evaluative and emotional dimension of mental wellbeing. Although using different measures of income and a younger age group, previous research indicated that income tends to influence the evaluative and the emotional dimensions differently [50]. Although specific reasons behind these differences lie outside of the scope of this article, it does appear that the welfare state effect is mediated by income for both dimensions also in the oldest old age group. Coping with one’s current income mediates the welfare state effect on the psychological functioning dimension, implying that that having a subjective sense of financial security impacts autonomy and a sense of accomplishment in oldest old age. A welfare state can support autonomy by ensuing that older adults have sufficient income to access what they need for a good life. 

A preference for ‘ageing in place’—that is, remaining living in the home environment as long as possible [51], has been found to foster independence and allow for higher levels of control and a higher quality of life [52]. A welfare state can facilitate ageing in place by ensuring adequate income, suitable housing options, access to services and social support within and outside of the home [53]. It is equally important to ensure autonomy within the context of residential care in terms of people’s decisions about their daily activities, maintaining dignity and upholding human rights [54,55]. Welfare state support is especially important in relation to high support needs. Financial deterrents are more likely in countries where individuals are expected to make their own contributions to institutional care, which tend to only take place only when it is absolutely necessary [56].

The mediation effect on the flow dimension illustrates how a subjective sense of financial security influences feelings of being interested, absorbed and enthusiastic. Longitudinal benefits of economic wellbeing in younger age groups indicates that higher income and/or increases in income are associated with higher levels of certain aspects of psychological wellbeing, such as purpose in life, self-acceptance, personal grown and environmental mastery, paralleling findings from the current study [57]. Ensuring access to leisure activities in oldest old age may be done by ensuring adequate transport and mobility [58] and access to day centres [59] or leisure activities [60].

The coping with income variable almost fully mediated the welfare state effect on the relational dimension suggesting that a subjective sense of financial security promotes a sense of being treated with respect, of being appreciated as well as providing and receiving help and support. The relational dimension was least affected by differing levels of welfare state support (see Figure 3). The link between having an adequate income and the relational dimension seems logical in the oldest old age considering that oldest old age is a time when support needs are likely to increase [9]. With less support from the welfare state, family relations and family financial support become critical, especially in the context of inadequate income levels. Intergenerational family support is more likely to be unbalanced and non-reciprocal in low-income contexts. Circumstances in which the older adult is predominantly the recipient of support from family members has been linked to lower levels of life satisfaction [61].

### 4.3. Place in Society and Mental Wellbeing

The positive association between status and subjective well-being in the general population is well-documented in previous research [62]. Subjective social status—that is, how people perceive their place in society—has been reported to have links to health and mental health outcomes [63] also in old age [64]. These associations have been found to persist even when statistically accounting for socioeconomic indicators, such as income, education and occupational prestige confirming the usefulness of the subjective measure [65]. Furthermore, cross country comparisons suggest that the health benefits related to high levels of subjective social status appears to be slightly more prominent in more affluent countries [65].

Welfare generosity (represented here by higher WSI values) aims to decrease poverty and increase equity by redistributing resources through social welfare and social protection. The current study showed a strong welfare state effect for the place in society variable, indicating that participants from countries with a higher WSI also reported a higher level of perceived placement in society. The place in society variable had a significant indirect effect on all mental wellbeing dimensions, implying a broad influence on the welfare state effect in this regard. A positive correlation between status and subjective wellbeing was previously reported in an age group under the age of 80 [66]. High levels of subjective social-economic status have also been reported in relation to mental wellbeing in older age groups, although less attention has been given to individual dimensions [67,68].

The current study found the place in society variable to explain part of the welfare state in terms of the evaluative dimension. Similarly, actions aiming to improve social status (for example by reducing social inequality) have been found to improve life satisfaction in the general population [69]. The current study also found place in society to influence the emotional dimension. Our findings parallel previous studies reporting the subjective position in society to be an important predictor of happiness in the general age group [24]. This was also found to be moderated by the welfare state, although to a lesser extent in the context of the Nordic countries compared to Eastern European countries [24].

The place in society variable also mediated the welfare state effect on the psychological functioning dimension in the current study. More equitable welfare states with higher levels of resources can be assumed to support the psychological functioning dimension of mental wellbeing. This is of particular relevance in older age as this age group tends to be ascribed low social status, and ageist attitudes are pervasive in many countries [70]. Ageism and associated negative health outcomes have been found to be more prevalent in countries with lower level of resources [71] and a high percentage of older adults [72].

The place in society item also mediated the welfare state effect on the flow dimension, which could potentially be stimulated by a variety of community activities, including voluntary or charity work, educational or training courses, sport, social or other kind of clubs and political or community-related organisations. All of these have been shown to follow a social gradient, with a higher social position increasing the likelihood of engagement in community activities among older people [73]. Studies also indicate that facilitating these actions impacts how engaged older adults are in these activities [74]. Furthermore, the place of society variable fully mediated the welfare state effect the relational dimension, highlighting an association between subjective social status and a sense of being treated with respect, feeling appreciated and being able to provide help and support.

The current study implies that life conditions in the daily environment has a mediating effect on many dimensions of mental wellbeing in oldest old age. Although three main variables were picked out in more detail, other variables also paint a similar picture. The variable ‘learn new things’ mediated the welfare state effect on all dimensions of mental wellbeing. Life-long learning is an area of limited attention in the oldest old age group. The results from this study could potentially provide a springboard for further study.

### 4.4. Protective Factors for Mental Wellbeing in Oldest Old Age; Some Policy Considerations

Building age-friendly environments may require increased attention to the smaller details in life. The World Health Organisation (WHO) highlights the need for facilitating access to activities, such as outdoor environments, social participation including community exchange and lifelong learning, social and neighbourhood cohesion and civic engagement also in old age [75]. By pinpointing the importance of different life conditions for mental wellbeing, the current study gains insight and support for policy actions, which develop mentally healthy environments for the oldest old. The results also highlight the need for a Mental Health in all Policies (MHiAP) [76] approach and the importance of involving multiple sectors, such as health, long-term care, transport, housing, labour and social services and actions on different levels, such as governmental and statutory services, civil society as well as individuals, communities and families [77,78].

The results from the current study also emphasise the multidimensional nature of mental wellbeing and emphasize the welfare state influence on different dimensions. Facilitating the opportunity to engage in actions that support mental wellbeing in all ages and circumstances needs to be driven by supportive policies. Interestingly, all three of the intervening variables discussed here centre around the notion of autonomy. Autonomy is an area highlighted in the MWIA, including enhancing control, facilitating participation and promoting inclusion (see Table A2). Support autonomy in oldest old age is an important endeavour for welfare states, considering that this age group is at increased risk of being excluded from both services and social contacts [79].

### 4.5. Strengths and Limitations

A strength of the study is its focus on an often neglected age group, which is an especially important area of attention considering the ongoing demographic transition. The current study accounts for the complex nature of both welfare state and mental wellbeing by a multidimensional approach using a composite measure of welfare state (WSI) and including five dimensions of mental wellbeing.

There are however some limitations to be taken into consideration. First, the cross-sectional nature of the ESS data increases the risk of cohort effects. ESS-data were collected from the community (as opposed to supported living environments) meaning there is a bias towards data from people with higher levels of mental wellbeing. Depending on the welfare state structure, individuals with higher health needs are generally more likely to live in institutional care, and it is important to recognise that these individuals are not included in the current data set. Subsequent analyses using longitudinal data could provide further information on cohort and age effects in relation to mental wellbeing and welfare state effects.

Furthermore, the data used in this study are relatively old, stemming from 2012, as this is the last point when the ESS module relating to Subjective and Social Wellbeing was included. It is worth acknowledging that both WS measures and their association to mental wellbeing at different ages may have changed since then. The stability of our results should be tested in later research also looking at longitudinal effects. It would also be useful to repeat these analyses should the ESS repeat the rotating modules used in this study.

It is important to recognise that various perspectives exist in terms of mediation analysis and that statistical models used to test mediation are simply predictive or descriptive and cannot be considered to be inherently causal [80]. In the case of the current study, we can only conclude that the intervening variables may have an influence in terms of the welfare state effect, but we cannot imply causality. Multiple mediators can be assumed to lie behind the welfare state effect.

## 5. Conclusions

The current study develops our understanding of how life conditions in the daily environment mediate the effect of the welfare state on mental wellbeing among the oldest old. The welfare state impacts on health are broad and multifaceted, and our study confirms that the higher the level of welfare state, the higher the levels of all mental wellbeing dimensions also among the oldest old. Although many of the intervening variables had a mediating effect on the different dimensions of mental wellbeing, subjective health, coping with income and place in society showed the highest mediating effects for the welfare state variable for all dimensions of mental wellbeing. For the psychological functioning and relational dimensions, this mediation effect was close to complete or was complete. All three variables centre around supporting aspects of autonomy.

## Figures and Tables

**Figure 1 ijerph-19-04363-f001:**
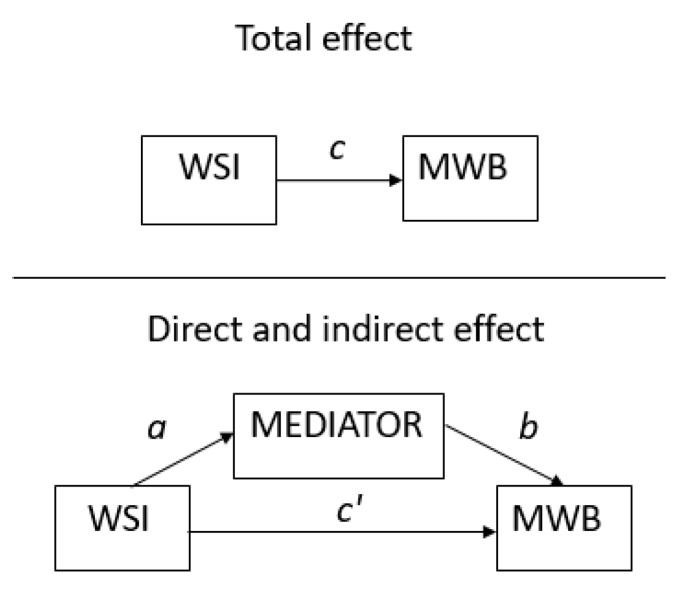
Mediation diagram denoting WSI effect on mental wellbeing with and without mediating aspects of WSI.

**Figure 2 ijerph-19-04363-f002:**
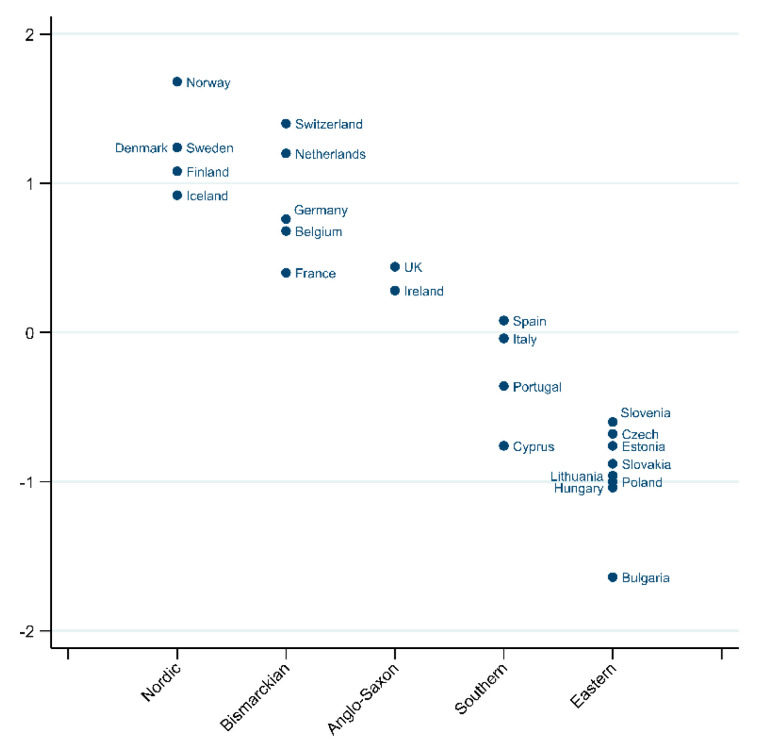
Country WS Index by WS Regime type for 24 European countries.

**Figure 3 ijerph-19-04363-f003:**
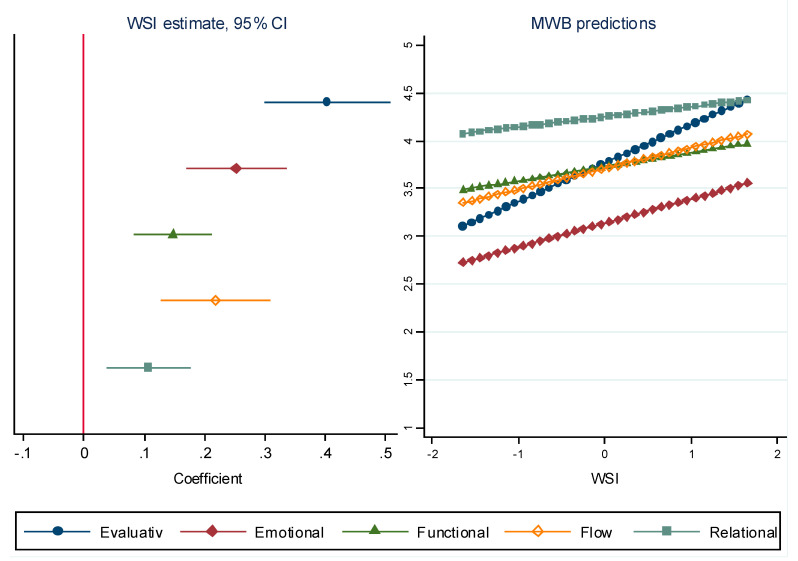
The total effect of WSI on mental wellbeing (coefficient and 95-percent confidence intervals) (**left panel**) and mental wellbeing-predictions for different values of WSI (**right panel**).

**Table 1 ijerph-19-04363-t001:** Mediator variables.

Intervening Variable ^a^	Mean	SD	CV ^b^	WSI Effect ^c^	N ^d^
Subjective general health (1–5)	3.1	0.93	0.31	0.411 ***	2053
Feeling about household’s income currently (1–4)	2.9	0.90	0.31	0.574 ***	2034
Hampered in daily activities by illness/disability/infirmity/mental problems (1–3)	2.2	0.75	0.34	0.114 *	2055
Take notice of and appreciate your surroundings (0–10)	7.2	2.22	0.31	0.155 **	1985
Most people can be trusted or you cannot be too careful (0–10)	5.0	2.64	0.53	0.336 ***	2044
Voted last national election (0–1)	0.8	0.42	0.55	0.183 ***	2048
Involved in work for voluntary or charitable organisations, how often past 12 months (1–6)	1.7	1.42	0.86	0.310 ***	2048
Take part in social activities compared to others of same age (1–5)	2.5	1.10	0.45	0.241 ***	1968
Place in society (0–10)	5.1	1.93	0.38	0.364 ***	1935
Learn new things in life (0–6)	2.8	1.88	0.66	0.266 ***	1996
Little chance to show how capable I am (1–5)	2.9	1.03	0.35	0.038	1951

^a^ Intervening variable from ESS, numbers in parentheses show the range of the variable with low values corresponding to low ratings. ^b^ CV = coefficient of variation= standard deviation/mean. ^c^ WSI effect in this table is for standardised values (mean = 0, standard deviation = 1) to be comparable between variables. ^d^ Missing values for items used to calculate the mental wellbeing measures implies a lower N than in the initial sample. Significance level denoted by stars as follows: *** *p* < 0.001, ** *p* < 0.01 and * *p* < 0.05. Means, standard deviations (SD), coefficient of variation (CV), coefficient for WSI effect on intervening variable (path a in Figure 1), and number of observations.

**Table 2 ijerph-19-04363-t002:** Mental wellbeing dimensions.

	Evaluative	Emotional	Functional	Flow	Relational
mean	3.80	3.16	3.76	3.75	4.27
SD	0.86	0.92	0.63	0.89	0.63
CV *	0.23	0.29	0.17	0.24	0.15
N **	2023	1969	1886	1928	1922

* CV = coefficient of variation = standard deviation/mean. ** Missing values for items used to calculate the mental wellbeing measures implies a lower N than in the initial sample. Means, standard deviations (SD), coefficient of variation (CV), and number of observations (N).

**Table 3 ijerph-19-04363-t003:** Coefficient for WSI without (top row) and with control for intervening variable *.

	Evaluative	Emotional	Functional	Flow	Relational
Total effect of WSI (c)	0.402 ***	0.252 ***	0.147 ***	0.218 ***	0.107 ***
Effect of WSI (c’) after control for:					
Subjective general health	0.273 *** (0.000)	0.092 *** (0.000)	0.056 (0.000)	0.095 * (0.000)	0.043 (0.000)
Coping with income	0.232 ***(0.000)	0.122 *** (0.000)	0.069 ** (0.000)	0.146 *** (0.000)	0.037 (0.000)
Hampered in daily life	0.380 *** (0.013)	0.219 *** (0.012)	0.128 *** (0.013)	0.193 *** (0.014)	0.095 ** (0.017)
Appreciate surroundings	0.350 *** (0.006)	0.203 *** (0.006)	0.110 *** (0.006)	0.143 ** (0.005)	0.060 *(0.006)
Trust in people	0.353 *** (0.000)	0.213 *** (0.000)	0.133 *** (0.010)	0.198 *** (0.015)	0.079 *** (0.000)
Voted	0.383 *** (0.001)	0.231 *** (0.001)	0.133 *** (0.001)	0.198 *** (0.001)	0.092 ** (0.002)
Voluntary work	0.368 *** (0.000)	0.211 *** (0.000)	0.116 *** (0.000)	0.181 *** (0.000)	0.089 ** (0.001)
Social activities	0.360 *** (0.000)	0.186 *** (0.000)	0.100 ** (0.000)	0.161 *** (0.000)	0.078 * (0.000)
Place in society	0.291 *** (0.000)	0.129 *** (0.000)	0.063 * (0.000)	0.100 ** (0.000)	0.032 (0.000)
Learn new things	0.348 *** (0.000)	0.181 *** (0.000)	0.095 ** (0.000)	0.132 * (0.000)	0.067 * (0.000)
Chance to show capabilities	0.402 *** (0.435)	0.236 *** (0.433)	0.138 *** (0.435)	0.199 *** (0. 434)	0.103 ** (0.435)

* Control for age and gender in all regressions. Significance level denoted by stars as follows: *** *p* < 0.001, ** *p* < 0.01 and * *p* < 0.05. Number in parentheses show the *p*-value for the mediator test (indirect effect).

## Data Availability

Not applicable.

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
