# Peer review of "Life Conditions as Mediators of Welfare State Effect on Mental Wellbeing among Oldest Old in Europe"

_ijerph, 2022, doi:10.3390/ijerph19074363_

Round 1

Reviewer 1 Report

A thoroughly referenced and researched analysis

Author Response

Many thanks for reviewing our article and thank you for the encouraging comments. We have revised the language as per your suggestion and hope the end result is more cohesive. 

Reviewer 2 Report

The paper is interesting and important. The only problem is carrying out analyzes, and therefore conclusions based on data from 10 years ago. The authors refer to it in the "limitation", nevertheless, due to the subject of the research, this approach should be more strongly justified in the introduction and methodology parts of this paper. Without that explanation conclusions and implications for practice (social policy) are of little use.

Author Response

Many thanks for reviewing our article and offering some advice on improvement.

We have carefully reviewed the language throughout and made some changes according to your suggestion. We hope this makes it more cohesive. 

In terms of the methodological issue with the data stemming from 2012:

We have clarified this approach further in the introduction from line 75 onward, in the paragraph beginning from line 98, and at the end of the manuscript beginning on line 462. The main reason behind using data from this year is that the ESS has not included the required wellbeing modules since 2012 making newer data not available.  It would most certainly be interesting to look at more recent data and are hoping ESS repeat the wellbeing modules in future collection waves. 

We also noted one erroneous variable was included in Table 3 and in Appendix table 2 - it has been removed. 

Reviewer 3 Report

Reviewer’s comments

This is a good paper , in my opinion. It focuses on our "oldest old", a population segment that has received less perhaps attention in research. The paper begins with a well-researched introduction/ background information and clearly stated objectives. The methodology is adequate, in my opinion and the findings are well reported. The discussion is also good.

I suggest authors consider adding one or two in-text citations to the following statements in the discussion. On page 9, the sentence from lines 334 to 336 says, "Although studies indicate that.... welfare state support appears to be... [49]". This sentence was supported by only one study, according to the authors. Since they mentioned that "studies indicate...", where are the other studies?

The authors also wrote on page 9, line 342, that "previous studies have  pinpointed.... [50]," yet they only mentioned one study. What additional research has been done in the past? Since they said "studies," the authors should consider adding more citations.

Overall, a strong paper. Thanks!

Author Response

Many thanks for reviewing our article and offering some advice on improvement.

We have carefully reviewed the language throughout and made some changes according to your suggestion. We hope this makes it more cohesive. 

The citation issues have now been clarified and addressed. We acknowledge that the text was indeed misleading in the original version of the manuscript. 

We also noted one erroneous variable was included in Table 3 and in Appendix table 2 - it has been removed.